# Effect of Enzymatic Pre-Treatment on Oat Flakes Protein Recovery and Properties

**DOI:** 10.3390/foods12050965

**Published:** 2023-02-24

**Authors:** Darius Sargautis, Tatjana Kince

**Affiliations:** Department of Food Technologies, Latvia University of Life Sciences and Technologies, Riga Street 22, LV-3004 Jelgava, Latvia

**Keywords:** oat protein concentrate, enzymatic hydrolysis, functional properties, amino acids

## Abstract

Oats are considered an exceptional source of high-quality protein. Protein isolation methods define their nutritional value and further applicability in food systems. The aim of this study was to recover the oat protein using a wet-fractioning method and investigate the protein functional properties and nutritional values among the processing streams. The oat protein was concentrated through enzymatic extraction, eliminating starch and non-starch polysaccharides (NSP), treating oat flakes with hydrolases, and reaching protein concentrations of up to about 86% in dry matter. The increased ionic strength from adding sodium chloride (NaCl) improved protein aggregation and resulted in increased protein recovery. Ionic changes improved protein recovery in provided methods by up to 24.8 % by weight. Amino acid (AA) profiles were determined in the obtained samples, and protein quality was compared with the required pattern of indispensable amino acids. Furthermore, functional properties of the oat protein, such as solubility, foamability, and liquid holding capacity, were investigated. The solubility of the oat protein was below 7 %; foamability averaged below 8%. The water and oil-holding reached a ratio of up to 3.0 and 2.1 for water and oil, respectively. Our findings suggest that oat protein could be a potential ingredient for food industries requiring a protein of high purity and nutritional value.

## 1. Introduction

Oat protein is famous for its nutritional value comprising one of the highest concentrations of essential amino acids among crops [1,2]. Oats are considered as being absent of carrying negligible amounts of potentially harmful protein, such as gluten [3,4], which is intrinsic to conventional crops, such as wheat or barley [5]. The protein amount is relatively high in oats and accounts for about 15–20% of the weight [6]; its main fraction by Osborne mainly comprises globulin [7,8]. Oats might substantially vary in protein content and amino acid profiles depending on their variety and growth conditions [9]. Protein location in crops determines their amino acid profile; protein vacuoles closer to the outer layers and fractioned as fiber are believed to display protein content that is richer in essential amino acids [10]. The superior nutritional properties of oat protein generate interest in utilizing it as a food ingredient [11], although a commercial protein concentration method has not yet been established.

Oat protein has been mainly concentrated through two general techniques, dry or wet fractioning, where the former provides the protein of lower purity, mainly due to its inability to eliminate the attached-to-protein crop fractions, in particular starch [12,13]. The most common wet-fractioning method proposes alkaline protein extraction with its subsequent precipitation at the isoelectric point [10,14]. However, the process of alkaline treatment apparently influences the structure of the protein at the very beginning of the process. Thus, the subsequent investigation of protein properties relies on the characteristics of the protein, which have already been predetermined by the initially applied extraction parameters. That might not replicate the properties of native oat proteins. In addition, the harsh alkaline treatment induces the formation of potentially harmful substances, such as lysinoalanine [15]. Reported modifications of processes, such as oat material defatting [16] or pre-treating oat material, in particular, oat brans with enzymes prior to alkaline extraction [17,18], facilitate or improve protein extraction, although limiting factors were not considered and still remain. Some functional properties of oat protein were recently reported, wherein the protein was extracted from oat brans treated enzymatically by glucoamylase [19]. The proposed method investigated the protein retained in the supernatant, which initially passed the relatively high centrifugal force. The functional properties of the protein were improved compared to the control, which passed the alkaline extraction. The protein recovery was not reported; however, it is believed that the recovery of the particle might be improved by changing the settling velocities, wherein an increased particle size might improve sedimentation [20], in particular the protein, finally resulting in increased protein recovery. Alternatively to the isoelectric precipitation discussed above, protein aggregation might be induced through an ionic shift [21]. Li and Xiong [22] recently reported on the effect of salt on oat protein aggregation. The ionic strength modified by ionizable salts, NaCl and NaP, increased the particle size of oat protein. We speculate that in the present study, the enzymatic protein extraction led by the successive increase in the size of the protein particle should result in the increased recovery of oat protein. In addition, the composition of the protein in the presence of salts alters due to the weakened electrostatic repulsion of the protein [22], causing the binding of protein subunits during aggregation formerly restricted to an association.

The objective of this study was to extract and concentrate oat protein by a method wherein oat starch and non-starch polysaccharides were subjected to enzymatic hydrolysis with subsequent protein concentration through separation. Furthermore, the influence of ionic change on protein aggregation and protein recovery was investigated; in addition, the amino acid redistribution and nutritional value among the processed streams were determined.

## 2. Materials and Methods

### 2.1. Materials

For each trial, commercial oat flakes (Latvia) were used: 17.6 g/100 g of protein in dry matter (DM), 5.7 g/100 g of fats in DM, 2.13 g/100 g of fiber in DM, 54.1 g/100 g of carbohydrates, 4.4 g/100 g of beta-glucans, and 0.01 g/100 g of salt, as sampled. Enzymes used for the hydrolysis of starch and non-starch polysaccharides were as follows: commercial-grade enzyme SQzyme HSAL as the source of alpha-amylase (HTAA; from Bacillus Licheniformis, alpha-amylase activity 40,000 u/mL; Suntaq International, Guangzhou, China), and complex commercial-grade enzyme Grainzyme FL with the main xylanase activity (XYL; from Trichoderma reesei, xylanase, beta-glucanase, and cellulase activities were 12,000 u/mL, 5000 u/mL, and 1000 u/mL, respectively; Suntaq International, Guangzhou, China). NaCl was a commercial-grade table salt; water deionized; NaOH 0.1 M; HCl 0.1 M.

### 2.2. Chemical Characterization Methods

The following methods were applied to characterize the samples: protein LVS EN ISO 20483:2014, moisture ISO 6496:1999, fiber ISO 5498:1981, fat ISO 6492:1999, and amino acids LVS EN ISO 13910-2005.

### 2.3. Protein Extraction from Oat Flakes

#### 2.3.1. Oat-Protein-Extraction Hydrolyzing Starch by Alpha Amylase

Oat flakes were mixed with previously heated water at the temperature of 80 ± 1 °C, wherein HTAA was added at the amount of 0.05% by volume. Then, while continuously stirring, oat flakes (room temperature) were added at the ratio of 1:10 by weight. The mixture was stirred periodically at intervals of about 30 s every 3 min using the hand mixer Promix (Phillips, Hungary) for 30 min while the temperature of hydrolysis was kept within the range of 75–80 °C. The hydrolysate was then separated by Hereus Multifuge X3 (Thermo Fisher Scientific, Osterode am Harz, Germany) at the G-force 900 for 1 s to separate the fiber. The obtained clarified hydrolysate was then separated at G-force 4800 for 5 min. The extracted protein biomass was washed with water at a ratio of 1 to 4 by weight. The washed protein biomass passed separation by the aforementioned centrifuge at G-force 4800 for 5 min and then dried in a 65 ± 2 °C hot air oven B5745-5-M (AEG, Germany) for 24 h. The dried oat protein was milled by hammer mill LM 3100 Perten Instruments (Perkin Elmer, USA) and a sieve of 0.8 mm. The separated fiber was dried in a 65 ± 2 °C hot air oven for 24 h. Obtained samples were coded as A1 for protein and AF1 for fiber.

#### 2.3.2. Oat-protein-Extraction Hydrolyzing Starch by HTAA and Non-Starch Polysaccharides by XYL

Oat flakes were mixed with previously heated water at a temperature of 60 ± 1 °C, wherein HTAA and XYL were each added at the amounts of 0.05% by volume. Room-temperature oat flakes were added to the water at a ratio of 1:10 by weight while the water was continuously stirred. The mixture was stirred periodically at intervals of about 30 s every 3 min using the hand mixer Promix (Phillips, Hungary) for 20 min. Then, the temperature of the mixture was raised to 80 ± 1 °C while keeping the stirring intervals at the same periodicity for the next 20 min. The subsequent processing steps were the same as the said above for the oat-protein-extraction hydrolyzing starch by alpha-amylase. The obtained samples were coded as AX1 for protein and AXF1 for fiber.

#### 2.3.3. Oat-Protein-Extraction Shifting-Ionic Strength of the Solution

Oat protein was extracted by the said-above methods, treating oat flakes with HTAA and a combination of HTAA and XYL prior to the separation step at G-force 4800. NaCl was then added to the clarified hydrolysate up to 0.1 M, then stirred using the hand mixer Promix (Phillips, Hungary) for 1 min and kept in a 75 °C hot air oven for 4 h. After the retention, the protein extraction was processed by the same methods as above, wherein HTAA and combined enzymes of HTAA and XYL were used. The obtained protein samples were named AR1 and AXR1 for the proteins, wherein HTAA was used only as an enzyme and in combination with XYL, respectively.

### 2.4. Protein Solubility

The samples obtained by the methods said above were subjected to protein solubility treatments in aqueous solutions wherein the pH levels were set at values of 3, 6, and 9. This method was adopted with minor modifications, as published by Morr et al. [23] and Sewada et al. [24]. Briefly, aliquots were prepared by dispersing 1 g of the obtained samples in a 0.1 NaCl solution. The dispersions were then set to a certain pH value by 0.1 N NaOH or HCl and adjusted to 50 mL of volume. The dispersions were continuously stirred using a magnetic stirrer for 2 h at room temperature. Measurements of pH were controlled by the pH-metre Mettler Toledo Seven Compact equipped with an Inlab Expert Pro-ISM pH-electrode. After 2 h, the samples were separated by means of centrifugation at G-force 4600 for 5 min with Hermle Z 206 A (Hermle Labortechnik GmbH, Wehingen, DE) at room temperature. The supernatants were filtered through the filtration paper FB-III-20 (GOST 12026-76, ash content, no more than 0.00133%, filtration capacity <26 s, bursting strength 5 kPa, Melior XXI, Ltd., Moscow, RU). The content of nitrogen in filtrates was determined using the Kjeldal method. The protein solubility index (*PSI*) for samples was determined according to Equation (1):(1)$$PSI=\frac{Protein\;in\;filtrate,\;\%\;\times weight\;of\;solution\;\left(g\right)}{Protein\;in\;dried\;sample,\;\%\;\times weight\;of\;sample\;\left(g\right)\;}\times 100\%,$$
the protein conversion factor determining oat protein is set to 6.25 × N.

### 2.5. Water and Oil-Holding Capacity

Water and oil-holding capacity were determined according to the method described by Mirmoghtadaie et al. [25] with minor modifications. The samples of protein concentrate were dispersed in deionized water or in refined deodorized sunflower oil at room temperature, then stirred by a vortex mixer VXHDDG (Ohaus, Parsippany, NJ, USA) for 1 min and kept for 30 min, periodically vortexing in periods of 10 min for 10 s. The vortex mixer speed for water and oil was set at 2500 rpm and 1200 rpm, respectively. The amount of substance subjected to measurement was 1 g per sample, with the water and oil ratio being 1:10 by weight. Dispersions after 30 min were centrifuged at G force 3000 for 5 min by centrifuge Hermle Z 206 A (Hermle Labortechnik GmbH, Wehingen, Germany) at room temperature. The supernatants of the samples were poured out, and the pellets were weighed. The water and oil binding capacities were expressed as the amounts of water and oil in grams retained per gram of sample of the protein concentrate.

### 2.6. Foaming

The method of determining the foaming capacity with minor modifications was applied as described by Mirmoghtadaie et al. [25]. Briefly, the samples of protein concentrate, in the amount of 1 g each, were dispersed in 33 mL of the deionized water and continuously stirred with the magnetic stirrer for a period of 30 min. Then, the dispersions were subjected to high shear mixing with a T10 Ultra Turrax (IKA Werke GmbH & Co. KG, Staufen, Germany), which lasted 5 min. The total volume of foam was measured at periods of 5, 10, 30, 60, and 120 min. The foaming capacity of the oat protein concentrate was calculated according to Equation (2):(2)$$Foaming\;capacity=\frac{Foam\;volume\;\left(\mathrm{mL}\right)}{Initial\;volume\;\left(\mathrm{mL}\right)}\times 100\%,$$

### 2.7. Data Processing

Friedman’s non-parametric test and Anova test were applied to analyze the median and mean differences, respectively, among the polar and non-polar amino acid groups previously validated by the Shapiro-Wilk normality test. Data in tables and graphs are expressed as the mean ± standard deviation for at least three replications, if not mentioned separately. Statistical analysis was conducted using R [26], and figures were produced using packages of ggstatsplot [27], ggplot2 [28], and Microsoft excel. RStudio [29] was used for Integrated Development Environment for R.

## 3. Results and Discussion

### 3.1. Protein Extraction

#### 3.1.1. Protein Recovery

Oat protein was extracted by eliminating the starch through enzymatic hydrolysis by treating the oat material with HTAA and XYL, affecting non-starch polysaccharides. Additionally, the step of the ionic shift was introduced to promote the aggregation of the protein. The results characterize the obtained samples’ protein concentration, crude oil, and fiber content, as well as protein yield among the obtained samples, as presented in Table 1.

Protein extraction was carried out in separate batches of about 400 g of initial oat material in each. The dried samples were analyzed for protein concentrations which varied from 84.6% to 86.5% in dry matter (DM) by weight, for A1 and AXR1, respectively. For example, it was reported that applying harsh alkaline protein solubilization allowed the concentration of the protein up to 68.4% in DM, wherein the pH of the slurry was set to 12.1 [10]. Some modified methods, including enzyme pre-treatment (treated with xylanase, alpha-amylase, glucoamylase, and cellulase), prior to the subsequent 2M NaOH alkaline extraction, resulted in a protein concentration of up to 82% [18] when oat brans were used as a raw material. Other reports demonstrated that oat brans, treated by means of amyloglucosidase, allowing the concentration of the protein up to 83.8%, although the yield was not reported [19]. Recently published methods, introducing protein-glutaminase with subsequent protein separation through ultrafiltration, revealed improved protein solubility, although the protein concentration was only achieved up to 52.4% [30].

In the current study, the concentration of the protein in fiber samples varied in the range from 30.3 g/100 g in AF1 to 39.6 g/100 g in AXF1, both in DM. The oil content was higher in samples wherein non-starch polysaccharides were affected by XYL enzymes. The retention step affected the oil concentration level in the samples of oat protein concentrate. The oil increased in both cases wherein the retention step with an ionic shift was introduced. The retention step substantially increased the protein yield. The increase in protein yield reached up to about 24.8% and 17.8% of raw protein in samples, wherein oat flakes were treated by HTAA and in addition to XYL, respectively. This can be explained by increased ionic strength in solutions, which finally promoted the development of protein aggregates due to a change in intramolecular electrostatic forces [31]. In addition, salts, in particular 0.1 M NaCl at neutral pH, might stimulate the insolubility of the protein. Such a phenomenon was linked to a declined electrostatic repulsion and the development of hydrophobic interactions [22]. Based on that, we concluded that a 0.1 M NaCl concentration stimulated the protein aggregation and facilitated its precipitation.

The difference in effect for samples treated with XYL should be considered in terms of the first separation of fiber at 900× *g*. The separation force was determined experimentally by the authors as minimal, effectively separating the insoluble fiber. Applying enzymes subsequently that break down non-starch polysaccharides promoted protein sedimentation along with the fiber. That apparently reduced the yield of the protein in AX1 compared to A1 free of XYL treatment. The high protein concentration in the fiber might also be considered as an associated material with aleurone and sub-aleurone layers which are typically rich in protein [6].

#### 3.1.2. AAs in Protein Concentrates

The amino acid compositions of the analyzed samples are given in Table 2. Generally, changes in the amino acid composition of products passing treatment might reveal their extent of modification [32]. In the present research, the AA profiles among the samples were relatively identical. Minor decreases of about 10% in Cys and about 8% in Met were observable in AR1 as compared to A1. On the other hand, slight increases of about 7% in His and about 8% in Ile were detected in AR1 as compared to A1. The ionic shift had no significant impact on the AA content in samples treated with XYL. The content of AAs in those samples was quite equal and varied in the range of 3% for certain AAs such as His and Ile. Other changes in the AA content between AX1 and AXR1 were even at a lower extent. 

Since the protein consists of AAs, its interaction with solvents, including water, assumes the involvement of functional groups or the peptide bonds of the individual AAs independently [33]. The presence of chaotropic salts influences electrostatic interactions with polar and charged groups, affecting hydrophobic interactions that enhance the unfolding tendency of the proteins [34]. Exposed groups are known to be important in salt-induced protein-protein interactions [35]. Given that the albumin fraction normally has an increased content of polar amino acids, whereas the increased amount of non-polar amino acids is typically attributed to oat globulins [36], AAs were grouped by polarity, although significant differences among the groups have not been revealed, as shown in Figure 1. The lack of significant changes in AA content might indicate that the ionic shift did not affect the protein composition in the protein concentrate.

In contrast to the similarity of AA profiles among the protein concentrates, the redistribution of AAs among the rolled oats and its derivatives obtained through the process, particularly the oat protein concentrates and oat fiber fractions, varied substantially and are discussed below.

#### 3.1.3. AA Redistribution among the Samples A1, AR1, AF1

AA redistribution was compared between the samples treated with HTAA, in particular A1, AF1, AR1, and the initial oat material; the percentage change is shown in Figure 2. A considerably increased concentration of Cys, Met, and Tyr was observable in protein concentrates. The peak of change was observed for Arg, of which the concentration substantially increased in the oat fiber stream. An increased amount of Arg in the bran fraction was also observed by Ma [10]. Arg prevails in 7S fractions of oat globulin [37], although this association might be considered with some degree of uncertainty assuming the challenging sedimentation of 7S at low-speed G-force, which was operated in the present study. Lys, which is considered a limiting AA in oat proteins, decreased in the concentrated protein, though it increased in the fiber stream. The decrease in Pro was observed in all samples treated with HTAA. Due to the ionic shift, the Cys and Met content was lower in AR1 than in A1, although AR1 contained higher amounts of His and Ile than A1.

#### 3.1.4. AA Redistribution among the AX1, AXR1, and AXF1

It is believed that some of the oat protein is bound within the non-starch polysaccharide matrix and could be effectively liberated by enzymes hydrolyzing non-starch polysaccharides [17], in particular glucosidases [38]. However, introducing XYL has not changed the AA profile in AX1 substantially compared to A1. Interestingly, the retention has not influenced AA redistribution; the resulting AA profiles of AXR1 and AX1 revealed negligible deviations only (see Figure 3). Generally, current AA redistribution harmonizes with the aforementioned profile, wherein oat flakes were treated with HTAA only. Lys content was reduced in protein concentrates, while it slightly increased in the fiber stream. Pro was present at a lower extent in all samples, whereas its concentration increase was observed in most of the analyzed AAs.

Apparently, the ionic shift does not promote the aggregation of various protein subunits; rather, the changed ionic charge stimulates protein aggregation, attracting proteins of the same fraction, or changes to forms disposed to faster sedimentation. On the other hand, the presence of xylanase during hydrolysis changes the concentration of AAs in the fiber stream considerably.

#### 3.1.5. Nutritional Value of Fractioned Oat

The amount of essential amino acids in all fractions was clearly higher than the FAO recommended for the ideal protein [39]. The composition of indispensable amino acids was surpassed in all samples. The graph representing averaged content (oat fractions averaged wherein A1, AR1, AX1, and AR1 are protein fractions, while AF1 and AXF1 are fiber fractions) of indispensable AAs in samples is represented by Figure 4. The only limiting amino acid in the samples was lysine, which is typically low in the initial oat material. A deficiency of lysine was observed in all samples. The summarized content of indispensable amino acids in averaged samples overcame the recommended 36% ratio for essential/total amino acid content [36]. The highest content of the summarized indispensable amino acids was determined for the oat protein concentrate.

### 3.2. Protein Solubility

The results of the solubility test for samples A1, AR1, and AX1 are presented in Figure 5. The solubility of the protein has reached about 6% in all ranges of pH values in the measured samples. Interestingly, the pH shift to the acidic or alkaline side improved the protein solubility only to an insignificant extent, despite the fact that the protein is susceptible to hydrolysis at harsh pH conditions [40], by its subsequent reduction in molecular size followed by an increased solubility [41]. Earlier reported solubility of the oat protein, obtained through alkaline solubilization with subsequent protein precipitation, ranged from 20% [42] to 70% [43]. Air-separated native oat proteins were reported as being soluble by more than 20% at pH 7 [44]. The solubility of oat proteins which passed enzymatic extraction was reported in the range of 10% to 50% at pH 9 and pH5, respectively [19]. However, it is rather difficult to explain such a low solubility of the current investigated samples as various factors might affect protein solubility, from salt concentration [8] to the oat variety used for tests [45]. Though, the authors of the present research speculated that the resulting data represented the protein self-assembling due to increased protein concentrations. It was observed that protein concentrations exceeding 1.0 mg/mL initiate protein aggregation, subsequently increasing the protein molecule in size through the association of proteins [46]. Increased protein molecule size might reduce protein solubility [41].

### 3.3. Foaming Properties

The foaming capacity of protein concentrate samples was evaluated and expressed as a change in the foam volume and stability within a 2-h period. Results are presented in Figure 6. Low protein concentration in the mixture was suggested to be used to avoid the viscosity effect on the colloidal stability [47]. The highest capacity was determined in AXR1 and A1, whereas the lowest capacity was found in AR1. The foam stability was poor for all tested samples and started to decrease sharply. The lowest foaming capacity and stability were observed for A1 as it was negligible at the start of the measurement and disappeared after 10 min. Interestingly, the ionic shift demonstrated dissimilar foaming characteristics; while the foaming capacity of AXR1 was the highest, the AR1, on the contrary, was the poorest. In general, the foaming capacity was substantially lower than reported by Kaukonen et al. [48], wherein the foaming capacity for the oat protein (protein content in the water extract used for the test was 0.33% with prior extracted lipids by CO_2_) reached up to 137% by volume. However, it should be considered that the molecular weight of the water-extracted protein was mostly detected at 10–15, 20–30, and 35–45 kDa bands. Those relatively smaller than the oat protein MW, and the methods used for extraction (water extraction), relate the reported protein to the soluble fraction, which was perhaps the fractions of water-soluble albumins. These results suggest that the foaming capacity of soluble oat proteins is higher than in proteins with limited solubility. A similar statement was reported earlier, indicating that albumins might contribute to foaming [49].

### 3.4. Water/Oil-Holding Capacity

The oil and water-holding capacities of oat protein concentrates are displayed in Figure 7. In general, the moisture retention, when operating with water, was practically identical for all samples. The highest and lowest water-holding capacity was detected for A1, wherein water was bound at ratios of 3.0 to 1. The lowest results were for AR1 and AX1, which ranged from a 2.84 and 2.83 to 1 ratio, respectively. Any substantial indifferences could be detected among the protein samples treated with salts or XYL enzymes, where the observed variation in the range of about 5% was too low to distinguish the influence of the presence of salt on the water holding capacity. The samples held oil at an average of about 2.19 to 1. Variation among the samples was negligible, with the highest value being for AX1 and the lowest for AR1, which was determined for samples at a 2.21 and 2.16 ratio, respectively. The extraction methods have not revealed a substantial impact on oil-holding capacity. The effect of oil and water holding was higher than reported by Mirmoghtadaie et al. [25], wherein the results for water and oil-holding capacity were determined at the ratio of 1.27 g/g and 1.73 g/g, for water and oil, respectively. The results were reported for the oat protein obtained through isoelectric precipitation after alkaline extraction. Similar alkaline extraction of the oat concentrate was also reported to produce a water-holding capacity ranging from 2.00 to 2.70 mL/g and an oil-holding capacity ranging from 2.25 to 2.80 mL/g [10]. Interestingly, the mentioned variation depended on the oat varieties. The enzymatically-extracted oat protein concentrate from brans had a reportedly higher water binding capacity of 3.73 mL/g and a lower oil-holding capacity which was determined in the range of 1.26 mL/g [19].

## 4. Summary

Oat protein, treated enzymatically, reached a concentration level of up to 86 g/100 g in DM. Breaking down non-starch polysaccharides by enzymes did not influence the protein concentration level, although this reduced the yield of protein extraction, supposedly due to the viscosity of the media reduction, which caused the increase in the speed of protein sedimentation. The ionic shift stimulated the aggregation of the protein, which resulted in a substantially increased rate of protein recovery. Breaking down the enzymatically non-starch oat polysaccharides did not affect the amino acid profile of the recovered protein; the composition of amino acids remained the same. Applying complex enzymes to depolymerize non-starch polysaccharides led to protein concentration in the fiber stream during fiber separation. The amino acid profile in the fiber and protein concentrate varied; the lysine content in the fiber was present to a higher extent. On the other hand, the concentration of the essential amino acids was higher in both analyzed streams, in the protein concentrate, and in fiber, compared to the initial material oat flakes. The liquid-enzymatic oat processing produced a practically insoluble oat protein concentrate. The water and oil-holding capacity of the protein concentrate was in the range of 2.8–3.0 g/g and about 2.2 g/g, respectively; no substantial difference was observed among the samples. The foaming capacity of the recovered oat protein concentrate was negligible. 

## Figures and Tables

**Figure 1 foods-12-00965-f001:**
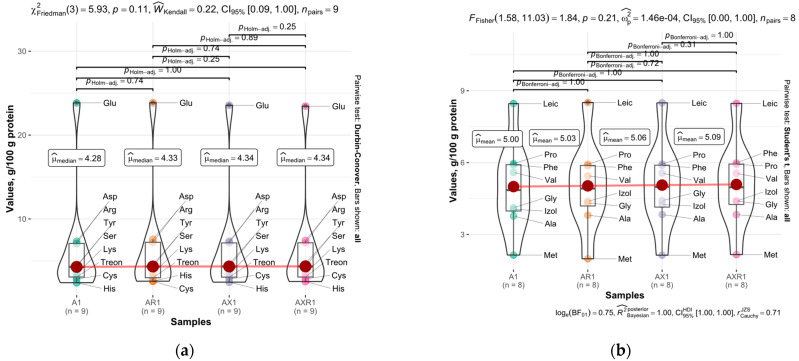
Amino acid redistribution in (**a**) polar and (**b**) non-polar groups among the samples A1, AR1, AX1, AXR1, values, mean, g/100 g protein. Processed data is plotted within the figures.

**Figure 2 foods-12-00965-f002:**
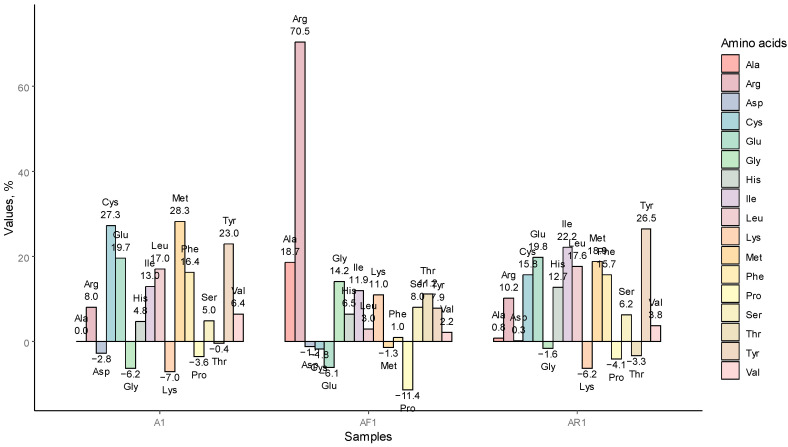
The percentage change of amino acid amounts in samples A1, AF1, and AR1, compared to initial oat material, oat flakes (FL1).

**Figure 3 foods-12-00965-f003:**
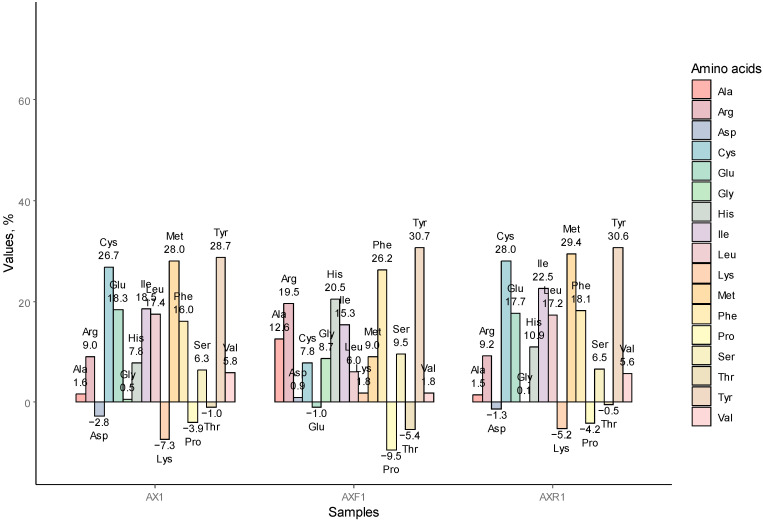
The percentage change of amino acid amount in samples AX1, AXF1, AXR1, compared to initial oat material, oat flakes (FL1).

**Figure 4 foods-12-00965-f004:**
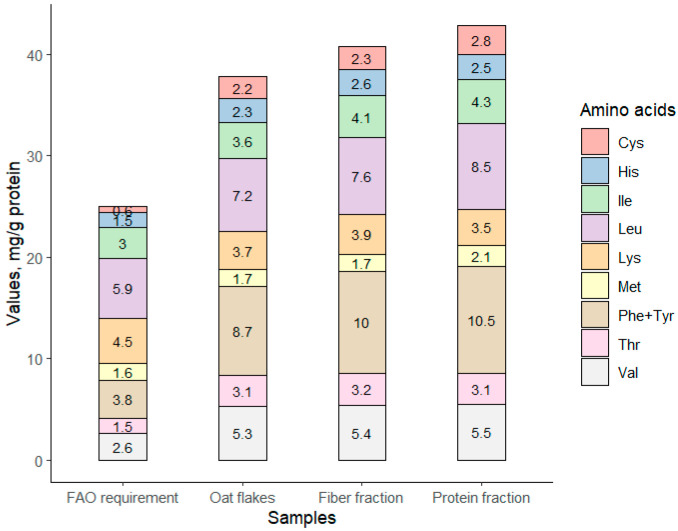
Amount of indispensable amino acids in initial oat flakes, averaged fiber and protein fractions, and FAO (2007) recommended values, mg/g protein.

**Figure 5 foods-12-00965-f005:**
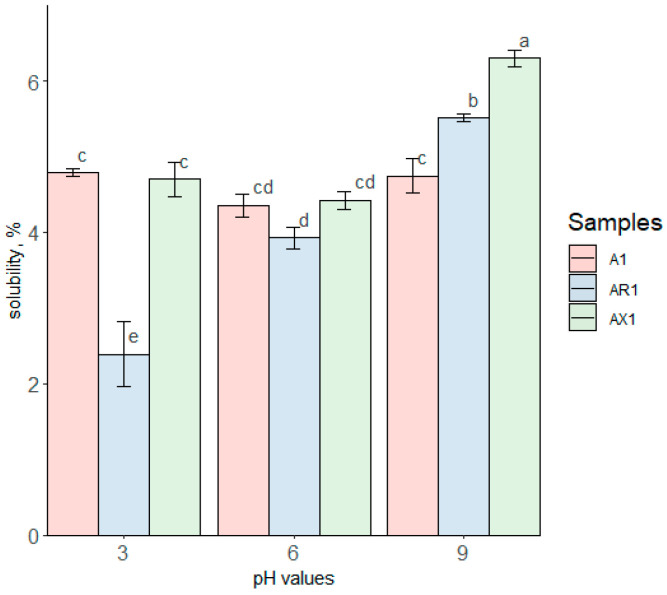
Protein solubility among the oat protein samples A1, AR1, and AX1 at pH 3, 6, and 9; %. Different letters indicate significant differences within the samples (*p* < 0.05).

**Figure 6 foods-12-00965-f006:**
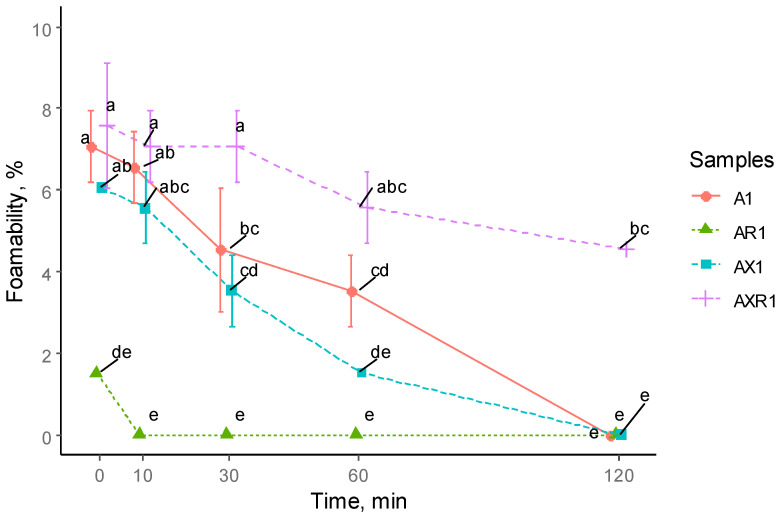
Foaming capacity of oat protein concentrates, %. Different letters indicate significant differences in each measurement (*p* < 0.05).

**Figure 7 foods-12-00965-f007:**
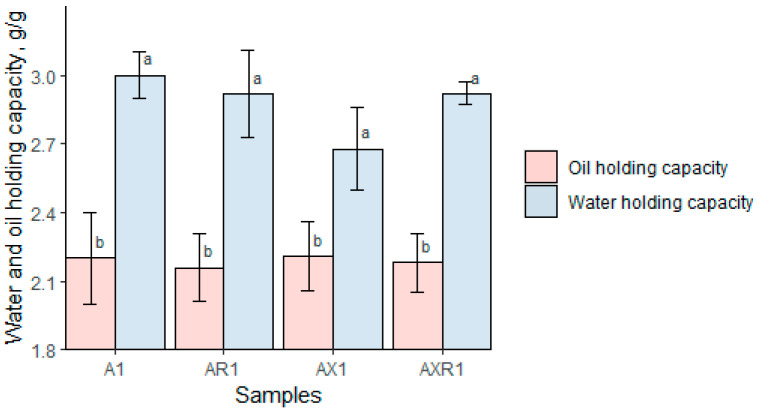
Water and oil-holding capacity ratios, g/g. Different letters indicate significant differences within the samples (*p* < 0.05).

**Table 1 foods-12-00965-t001:** Chemical characterization of oat protein concentrates, g/100 g in DM. Different letters within a column indicate significant differences for each parameter (*p* < 0.05).

Samples	Protein	Crude Oil	Fiber	Protein Yield *
FL1	17.56 ± 0.03 d	5.7 ± 0.11 d	2.1 ± 0.1 b	NA
A1	84.64 ± 1.64 a	3.0 ± 0.14 e	1.4 ± 0.08 c	35.9 ± 0.70 c
AR1	84.2 ± 1.89 a	5.2 ± 0.06 d	NA	44.8 ± 0.83 b
AX1	85.86 ± 1.80 a	5.7 ± 0.08 d	1.1 ± 0.05 c	28.1 ± 0.56 e
AXR1	86.46 ± 2.23 a	6.3 ± 0.20 c	NA	33.1 ± 0.51 d
AF1	30.30 ± 0.62 c	7.6 ± 0.16 b	5.5 ± 0.37 a	35.9 ± 0.73 c
AXF1	39.36 ± 0.76 b	9.0 ± 0.40 a	5.2 ± 0.34 a	47.6 ± 1.02 a

* % of protein content in initial material.

**Table 2 foods-12-00965-t002:** The amino acid compositions of oat protein in analyzed samples, with g/100 g of total protein. Means followed by the same letter within a row indicate no significant difference among the samples (*p* < 0.05).

FL1	A1	AR1	AX1	AXR1	AF1	AXF1	FL1
Ala	3.77 ± 0.13 b	3.77 ± 0.05 b	3.8 ± 0.1 b	3.83 ± 0.05 b	3.83 ± 0.05 b	4.48 ± 0.14 a	4.25 ± 0.11 a
Arg	6.55 ± 0.17 d	7.08 ± 0.19 cd	7.22 ± 0.08 c	7.14 ± 0.16 c	7.16 ± 0.34 c	11.17 ± 0.21 a	7.83 ± 0.15 b
Asp	7.54 ± 0.28 a	7.33 ± 0.33 a	7.57 ± 0.2 a	7.33 ± 0.17 a	7.44 ± 0.2 a	7.45 ± 0.17 a	7.61 ± 0.32 a
Cys	2.23 ± 0.05 c	2.83 ± 0.12 a	2.58 ± 0.08 b	2.82 ± 0.03 a	2.85 ± 0.03 a	2.18 ± 0.09 c	2.4 ± 0.12 bc
Phe	5.07 ± 0.18 c	5.9 ± 0.08 b	5.86 ± 0.22 b	5.88 ± 0.06 b	5.99 ± 0.07 ab	5.12 ± 0.25 c	6.4 ± 0.17 a
Gly	4.39 ± 0.14 bc	4.12 ± 0.14 c	4.32 ± 0.19 c	4.41 ± 0.19 bc	4.39 ± 0.12 bc	5.01 ± 0.15 a	4.77 ± 0.08 ab
Glu	19.91 ± 0.42 b	23.84 ± 0.65 a	23.85 ± 0.79 a	23.55 ± 0.31 a	23.44 ± 0.47 a	18.7 ± 0.3 b	19.71 ± 0.76 b
His	2.29 ± 0.05 c	2.4 ± 0.04 bc	2.58 ± 0.12 ab	2.46 ± 0.07 bc	2.54 ± 0.05 b	2.44 ± 0.1 bc	2.76 ± 0.08 a
Ile	3.59 ± 0.16 c	4.05 ± 0.06 b	4.38 ± 0.14 a	4.25 ± 0.13 ab	4.39 ± 0.06 a	4.01 ± 0.06 b	4.14 ± 0.15 ab
Leu	7.23 ± 0.13 b	8.46 ± 0.17 a	8.51 ± 0.19 a	8.49 ± 0.1 a	8.47 ± 0.36 a	7.45 ± 0.3 b	7.66 ± 0.17 b
Lys	3.71 ± 0.05 bc	3.45 ± 0.08 d	3.48 ± 0.06 cd	3.44 ± 0.05 d	3.51 ± 0.16 cd	4.12 ± 0.04 a	3.78 ± 0.11 b
Met	1.67 ± 0.08 c	2.14 ± 0.02 a	1.99 ± 0.04 ab	2.14 ± 0.08 a	2.16 ± 0.06 a	1.65 ± 0.08 c	1.82 ± 0.05 bc
Pro	6.18 ± 0.14 a	5.96 ± 0.15 ab	5.93 ± 0.26 ab	5.94 ± 0.25 ab	5.93 ± 0.11 ab	5.48 ± 0.25 b	5.6 ± 0.16 b
Ser	4.08 ± 0.05 a	4.28 ± 0.17 a	4.33 ± 0.15 a	4.34 ± 0.13 a	4.34 ± 0.17 a	4.41 ± 0.13 a	4.47 ± 0.2 a
Tyr	3.59 ± 0.12 b	4.41 ± 0.2 a	4.54 ± 0.12 a	4.61 ± 0.2 a	4.68 ± 0.07 a	3.87 ± 0.12 b	4.69 ± 0.16 a
Thr	3.09 ± 0.08 b	3.08 ± 0.14 b	2.99 ± 0.12 b	3.06 ± 0.1 b	3.08 ± 0.05 b	3.44 ± 0.07 a	2.92 ± 0.04 b
Val	5.25 ± 0.26 a	5.59 ± 0.11 a	5.45 ± 0.08 a	5.56 ± 0.2 a	5.55 ± 0.21 a	5.37 ± 0.17 a	5.35 ± 0.07 a

## Data Availability

Data is contained within the article.

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
