# Peer review of "Effect of Enzymatic Pre-Treatment on Oat Flakes Protein Recovery and Properties"

_foods, 2023, doi:10.3390/foods12050965_

Round 1

Reviewer 1 Report

The subject of the manuscript and the scope of research are scientifically very interesting and constitute an important contribution to the knowledge in the field of obtaining protein from oat flakes. Oats are considered to be an exceptional source of high-quality protein, and methods of obtaining protein need to be improved.

The brevity of the entire publication should be emphasized, a wide and relevant research scope was considered, and scientific interpretation and discussion of the results was applied.

The novelty of the research was the extraction and concentration of oat protein using enzymatic hydrolysis of starch and non-starch polysaccharides. In addition, a small addition of NaCl salt was used to study the effect of ion transformations on protein aggregation and recovery. Although the methods used did not significantly affect the results of the research, the described issues are an important contribution to knowledge in this area.

The disadvantages of the manuscript are the need to carefully check the text because some sentences need to be corrected, e.g. lines 29, 235 (unnecessary small font), 274. Some drawings are hard to read, the font is too small or the text overlaps the graph. Too little information in the caption of the figure, maybe it's worth adding information about the encoding method. Axes in graphs should have an improved scale range and less gradation.

“the influence of ionic change to protein aggregation and protein recovery” “Oat protein extraction shifting ionic strength of the solution”

The above statements regarding the use of NaCl should be improved and more unified, they also lead to a different way of interpreting such a treatment (e.g. the action of the electromagnetic field ...?) instead of simply adding NaCl salts. However, consideration may be given to adding this point to the title of the manuscript.

“Our findings confirm the oat protein has a great potential in food industries requiring protein of high purity and nutritional value.” - The last sentence in the abstract does not match the content of the manuscript. The abstract and conclusions can be slightly improved.

I suggest you consider using the phrases "toxic protein, such as gluten" and "toxic substances, such as lysinoalanine.

Author Response

Good afternoon,

Thank you very much for highly valuably review. The responses are attached. The responses are highlighted in red.

Reviewer 2 Report

The manuscript is well-written. However, the discussion parts need to be improved. 

-Discussion shall be written in a more didactic manner. Other than discussing whether the findings supported the expectations you had or your hypothesis, you also need to contextualize the results within the existing theory or research and consider alternative explanations while arguing your stance. 

-Standardize and rearrange the position of the superscripts in the figures.

-In 2.2 Chemical characterization methods, only the method numbers are given without description of the methodology. Please elaborate the methods used including the sample preparation steps.

-Reference no. 15 self-citation detected. Please cite the source of the information appropriately.

Author Response

Good afternoon, 

Thank you for highly valuable comments. Please find responses highlighted in red attached. 

Reviewer 3 Report

Dear authors,

The manuscript entitled “Effect of enzymatic pre-treatment on oat flakes protein recovery and properties” is an interesting topic that could be of interest for readers. I have only one observation.

Materials and methods:

Line 77. The meaning of the acronyms should be added (HSAL and HTAA)

Results and discussions:

Lines 173-174. This sentence seems from the materials and methods section

Line 211. I consider that "table 2" should be written with a capital "T"

Line 241. What is the explanation for the increase in Cys, Met and Tyr?

Author Response

(The authors gave the same response as above.)

Reviewer 4 Report

Please revise the manuscript upto greater extent. 

The manuscript entitled “Effect of enzymatic pre-treatment on oat flakes protein recovery and properties” contains an important study. I have some comments to revise the manuscript.

Abstract:

Line 15-16: The sentence structure is very confusing. Please revise it.

Line 17: foamability, liquid holding capacity should be “foamability and liquid holding capacity”

Line 18: The solubility of oat protein was below 7 %; foamability averaged below 8%. These values seems very low. Please check further.

Line 19: “Our findings confirm the oat protein has” should be “Our findings confirm that the oat protein has”

Line 21: protein functional properties. I think it’s better to use only functional properties

Introduction:

Line 40-44: This is a big sentence. Please revise it.

Line 60: leaded should be led, I think?

Line 65: this this study??

Is this method of protein extraction is novel? You should include in introduction few other materials for which this method has been effectively used if not novel? Introduction needs to be improved with more literature studies.

Materials and methods

Line 94-95: The hydrolysate was then separated by Hereus Multifuge X3 (Thermo Fisher Scientific, Germany) at the G-force 900 for 1 second to separate fiber?? How did you use one second time for centrifugation?

Line 167-168: Please remove the extra space.

Why did not you calculate other functional properties like foam stability, emulsion properties, structural properties of protein concentrate?

Results and discussion

“Discussions” is always “discussion”

Line 211: make the table 2 as Table 2.

Line 223: (see Figure 1) please correct it to as shown in Figure 1.

Line 242: which concentration substantially, it should be corrected?

Line 290: reduce in molecular size should be ‘reduction in molecular size’

Line 328: were summarized in Figure 7. It should be corrected as “are displayed in Figure 7”

Line 334: the influence the presence of salt?????? Should be the influence of the presence of salt.

What would be the application of these proteins because they have very low functional properties.

Summary:

Line 360: leaded to protein concentration in the fiber stream. It should be corrected as “led to protein concentration in the fiber stream”

Overall comments:

  1. The manuscript must be Grammarly checked.

Author Response

(The authors gave the same response as above.)

Round 2

Reviewer 4 Report

The manuscript is suitable for publication

Author Response

Thank you for revision.